# Dealing with radiation risks in systemic cancer treatment: Perspectives of practitioners and patients in French hospitals

**Solenn Thircuir**[1]*, **Héloïse Pillayre**[2], **Johannes Starkbaum**[3], **Erich Griessler**[3]

**1** ESA Business School, Beirut, Lebanon, **2** Chaire Santé, Sciences Po Paris, Paris, France, **3** Institute for Advanced Studies, Vienna, Austria

* solenn.thircuir@live.fr

**Editor:** Hesham M.H. Zakaly, Ural Federal University named after the first President of Russia B N Yeltsin Institute of Physics and Technology: Ural'skij federal'nyj universitet imeni pervogo Prezidenta Rossii B N El'cina Fiziko-tehnologiceskij institut, RUSSIAN FEDERATION

## Abstract

Systemic radionuclide therapy (SRT) using substances such as $^{177}$Lu is an approach in cancer treatment that aims to destroy malign tissues by injecting radionuclides directly into patients' bodies via the bloodstream. This treatment connects benefits of care with risks related to radioactivity. Our research conducted in French hospitals shows that managing risk is an integral part of SRT, spanning from implementation, hospitals' protocols, specific management, hospital settings, and training, to the individual experiences of health professionals and patients who are both exposed to radioactivity. This article argues that understanding how risks are managed in SRT not only requires making them identifiable, quantifiable, and calculable through medical devices in the context of evidence-based medicine, but also necessitates fostering trust throughout the treatment. This article explores and provides insights into three intertwined dimensions of trust in risk management: epistemic, (inter)-organizational, and interpersonal.

## Introduction

During the last twenty years, innovative therapeutic strategies [1] have drastically increased the effectiveness of cancer treatment and led towards what medical actors call "personalized" or "precision medicine" [2]. $^{177}$Lu is increasingly applied in systemic radionuclide therapy (SRT) and is said to allow a reduction in radiation doses while increasing the quality of results [3,4]. It sparks hopes for considerably improving patients' health outcomes and overall quality of care. However, besides the expected benefits, the use of radioactivity in medicine in general, and the innovative method in particular, also raise issues of risk, regulation, acceptance and normalization.

Neuroendocrine tumors (NETs) are rare and most commonly affect the gastrointestinal tract, lungs, and pancreas [5]. Their conventional treatment involves, e.g., analogs of somatostatin, chemotherapy, or other radionuclide therapies. $^{177}$Lu is indicated for a group of patients that have been diagnosed with NETs. It appears to be a promising therapy in some cases as it can improve patients' health outcomes, slow down the disease's progression, increase patients' life expectancy [6], and decrease some side effects of cancer therapy. The use of medical imaging for this treatment plays a central role in supporting clinical decision-making, developing prediction models, and therefore, integrating information that can assess

**Data availability statement:** Data cannot be shared publicly because of sensitive information regarding patients interviewed. Data are available from the Institutional Review Board of the Institute for Advanced Studies for researchers who meet the criteria for access to confidential data. Data request can be made to Dr. Erich Griessler, Techno-Science and Societal Transformation Research Group, Institute for Advanced Studies (IHS), Josefstädterstraße 39, 1080 Vienna, +43 1 59991 170; erich. griessler@ihs.ac.at

**Funding:** This project was funded by the Austrian Science Foundation (FWF) within the ERA-NET PerMed Programme (Grant-DOI 10.55776/I4732) from 01.05.2020 to 31.10.2023. Information about the grant and the project is available on the FWF website (https://www.fwf.ac.at/forschungsra-dar/10.55776/I4732) and the project website (https://popeye.upatras.gr/). The funders had no role in study design, data collection and analysis, decision to publish, or preparation of the manuscript.

**Competing interests:** The authors have declared that no competing interests exist.

the risk of specific tumor outcomes [7]. Scientific contributions that address Ethical, Legal, and Social Aspects (ELSA) related to SRT and [177]LU so far focus primarily on radiation risks and related questions of dosimetry [8,9], the institutional embedding of this treatment [10], as well as on general ethical debates, which provide limited guidance for clinical practice [11]. While some authors [12], emphasize the need to involve diverse stakeholders, the knowledge of patients' perceptions of the treatment is limited. Some studies provide insights on NET patients more generally [13] but there hardly are studies that link these perspectives to SRT.

[177]Lu, Lutathera, or lutetium Lu 177 Dotatate are synonyms designated to the treatment observed in this article. It has slowly been implemented across Europe since the demonstrated success of NETTER-1, a pivotal, phase 3, randomized, multicenter, open-label, active-control trial [14]. Following the European Medicines Agency (EMA) approval in 2017, [177]Lu was made available in the European Union starting with Germany, Italy, France, the United Kingdom and the Netherlands.

During this treatment, the radioactive substance is used for diagnostic and therapeutic purposes. The treatment is considered highly attractive to patients because it is said to have a significant impact on tumor cells with fewer side effects than other therapies, thus emphasizing a significant diminution of risks [15]. During [177]Lu treatment, patients receive four cycles of radio nuclear injections at eight-week intervals to target tumor cells. Each is followed by Positron emission tomography (PET) imaging to monitor the treatment's efficacy while ensuring that side effects are limited. In brachytherapy, i.e., internal radiation therapy, patients are hospitalized and follow a treatment protocol to reduce contamination risk to themselves and others. They have follow-ups for a few months. Imaging techniques evaluate the state of health of the patients as well as the efficiency of the treatment, thus also minimizing risks.

Not all NET patients are eligible for [177]Lu: individual parameters, such as age and pathophysiological status, play a role in patient selection. Eligible patients suffer from inoperable tumors, either metastatic or locally advanced, growing progressively, whose primary site is midgut with low or intermediate grade, and well differentiated. Also, they need to show the slow progression of the tumor. Those receiving the treatment have to be adults and must have already passed different lines of other treatment. Patients with pre-existing risk factors, such as diabetes, pregnancy, or compromised renal function, are not eligible. Multidisciplinary tumor boards usually select patients after reviewing risks and benefits [8].

Fig 1 illustrates the pipeline for [177]Lu systemic radionuclide therapy (SRT) and the roles of various healthcare professionals involved at each stage of the treatment process. The pipeline begins with patient selection, undertaken by a multidisciplinary tumor board, which assesses eligibility based on criteria such as tumor progression, overall health, and previous treatments. Once patients are selected, dosimetry and initial imaging are performed by medical physicists and nuclear physicians to establish a personalized dose plan, ensuring that radiation exposure is both effective and safe. In the preparation and administration phase, radio pharmacists and radiology technicians are responsible for the handling and safe administration of the radiopharmaceutical, adhering to strict radiation safety protocols. Following injection, nuclear physicians and nursing staff monitor the patient's response and oversee radiation safety, a task supported by radioprotective advisors who ensure all necessary precautions are maintained. Finally, follow-up imaging and patient management are conducted to evaluate treatment efficacy and maintain ongoing safety; this stage involves medical imaging professionals and includes consultations with nuclear physicians and oncologists. Figure 1 underscores the integrated, collaborative roles that healthcare professionals play in managing both the therapeutic and radiological aspects of [177]Lu treatment, reflecting a comprehensive approach to risk management throughout the SRT process.

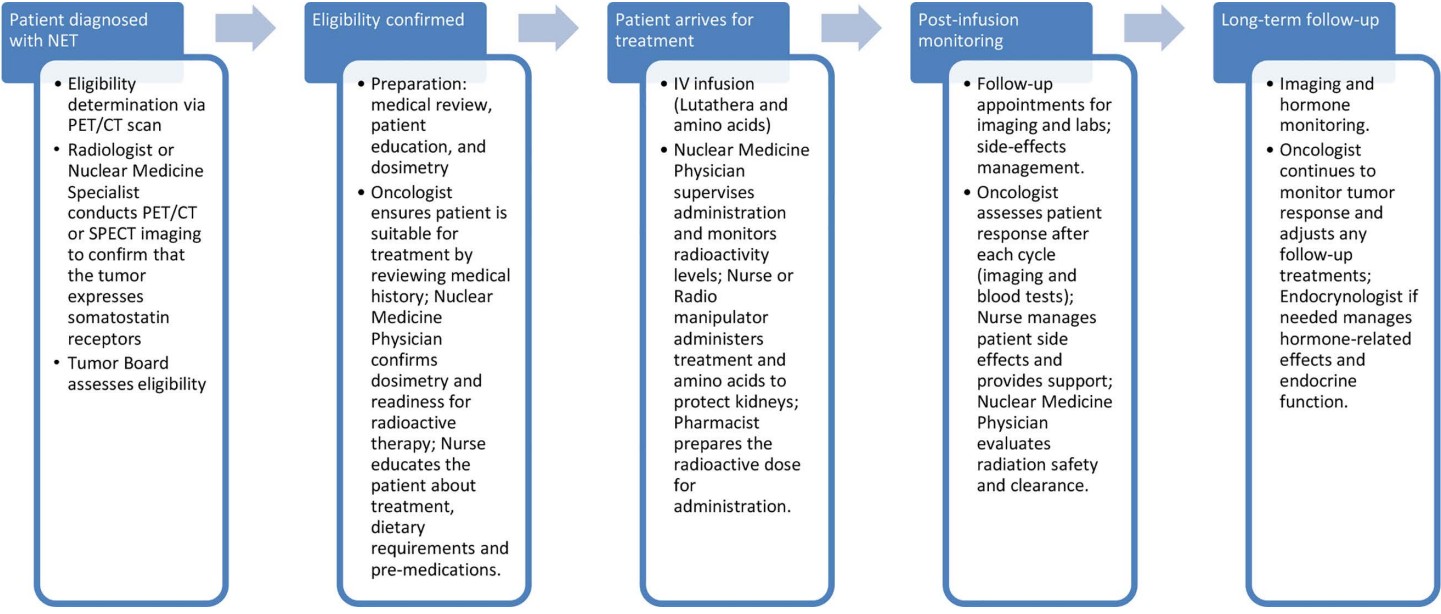

**Fig 1.** [177]Lu **Treatment Pipeline and Involvement of Healthcare Professionals.**

Sociologists distinguish *risk* from *uncertainty*, with the former being characterized by its calculability [16]. Since the 18th century, a probability-based way of conceptualizing hazards has been at the heart of medicine, shaping diagnosis, therapy, and more generally, clinical judgment [17]. Although numerous studies from social science focus on the benefit side of innovative treatments, several works have shown that risk assessment plays a key role in technology acceptance [18].

Medical treatments using radioactivity raise questions of risks and trust in medical innovation. Indeed, radioactivity recalls environmental and health disasters such as Chernobyl that left their mark in collective memory [19,20]. However, radioactivity has been utilized for a long time for medical treatment [21]. According to the American Cancer Society, the medical use of radioactivity for diagnosis and therapy started at the turn of the 19th century. However, at the same time, scientists discovered that radiation could both cause and cure cancer. The use of radioactive components to treat patients within nuclear medicine departments shapes the coexistence of radiation protection and care in dealing with risks [22]. These patients face different types of risks related to the stage of the treatment. Identifying the risks entangled with such advanced treatments has been essential for patients and health professionals. Our research shows that their acceptance to radiation exposure, and to agree to this risk, relies on the capacities of hospitals and health professionals to regulate and control radioactivity and to convey confidence in these actions.

The application of radioactivity in medicine is regulated on national and international levels. The first step for implementation in hospitals is getting approval from national authorities. The manufacturer, Advanced Accelerator Applications, a subsidiary of Novartis Group, received the first European Medicines Agency approval in 2017, and Food and Drug Administration (FDA) approval in 2018[4]. The treatment also received an Orphan Drug designation. This designation provides incentives to assist and encourage the development of drugs for rare diseases. The manufacturer became the supplier of this drug across Europe which ensures consistency in the product's quality, formulation, and safety standards. The treatment protocol for [177]Lu is in general consistent across Europe.

In France, according to the Nuclear Energy Agency (Autorité de Sureté Nucléaire, ASN), radiological protection is a term "*applied to the protection of workers, patients, the public and the environment from the harmful effects of exposure to ionizing radiation, and the means for achieving this*".[5] Health professionals also have to follow the guidance provided by the manufacturer based on the Nuclear Regulatory Commission which is an independent agency of the United States government tasked with protecting public health and safety related to nuclear energy. Health professionals also have to follow the hospital's protocols, and they are instructed to "*keep radiation as low as reasonably achievable*" [23].

This paper addresses the gap in empirical research into the clinical translation of SRT in general and [177]LU in particular. By taking a sociological perspective and applying qualitative methods of empirical research it investigates the implementation processes of [177]LU in French hospital settings and what they mean for health care professionals and patients. Applying the concepts of risk and trust, it will show how both are generated in an interplay of working processes, medical and other devices, structured spaces, and communication between the actors involved. The paper will show how trust arises from this articulation and lays the foundation for implementing the successful treatment. Only if the hazards of radiation can be "dealt with", i.e., turned from uncertainty into risks, trust in the method can be generated such that the treatment can be implemented.

The first part of the paper will show how radiation hazards are turned from uncertainty into calculable risks by focusing their specific management. The second part looks at how the new therapy impacts the organization of healthcare and the relationship between physicians, other health care professionals and patients. In a third part, by focusing on [177]LU, the paper will show how sentiments of security and trust are mobilized by social processes and interaction to implement an innovative treatment into existing hospital socio-spatial environments. We show that the creation of safety and trust are mutually dependent on one another and are based on social, political, organizational, and emotional elements that are created and reproduced within the relationship between medical institutions, staff and patients. The paper will finally conceptualize risk in this context by exploring how the technological dimension of the treatment is just one element in understand how hazards related to radiotherapy are managed and turned into risks that can partly be controlled.

## Methods

This paper is based on research conducted in the interdisciplinary European project POPEYE: Personalized optimization of prognostic and therapeutic protocols with Lu-177 for MNETs through the development of advanced computational tools and a portable detection system. Empirical research in the social science part of POPEYE started with exploratory interviews with consortium members. The main empirical data consists of twenty-nine qualitative semi-structured interviews as well as two ethnographic observations of the treatment injection carried out in 2020/21. Interviews with patients and health care professionals from each of the professions involved in the treatment were conducted in nine French hospitals. The sampling of the interviews allowed a broad insight into the multidisciplinary, and patient perspective on the treatment and risk related questions in different French clinical organizations applying [177]Lu (c.f. Table 1).

Interviews with health care professionals lasted about one hour each. In addition, five patients were interviewed who received treatment in Lyon, Bordeaux and Toulouse. Finally, one representative of the French patient self-help organization APTED (Association of Patients with Various Endocrine Tumors) was interviewed. The patients, three of them being female and two being male, were between 58 and 82 years old. One patient was interviewed a second time a few months after the end of the treatment on his initiative. The interviews

**Table 1. Distribution of interviewees by occupation and gender.**

| Occupation | Number of interviews | Women | Men |
|---|---|---|---|
| Nuclear physicians | 6 | 3 | 3 |
| Radio pharmacists | 3 | 2 | 1 |
| Radio manipulators | 2 | 1 | 1 |
| Medical physicians | 2 | 1 | 1 |
| Endocrinologists | 2 | 1 | 1 |
| Gastroenterologists | 2 | 0 | 2 |
| Team manager | 1 | 0 | 1 |
| Radio protection advisors | 2 | 1 | 1 |
| Nurse | 1 | 1 | 0 |
| Nursing Assistant | 1 | 1 | 0 |
| Translational research engineer | 1 | 0 | 1 |
| Patients | 5 | 3 | 2 |
| Patient representative | 1 | 1 | 0 |
| Totals | 29 | 15 | 14 |

with patients were made face to face and lasted between approx. half an hour to one and a half hours each.

In interviews, health professionals were asked about their career and current activity, the challenges and impacts of the treatment, health economics and technical aspects of the treatment, as well as their opinion on the personalized approach of it. Patients were asked to share their medical history, how they became eligible for the treatment, their perception of the treatment experience, as well as potential improvements.

Access to interviewees was gained through "snowball sampling" [24]. Health care professionals were asked after interviews for contacts to recruit further interviewees. Nuclear medicine physicians who accompanied the treatment also put the team in contact with patients who volunteered to take part in interviews. Some interview partners took initiative and contacted the researcher after a presentation of the POPEYE project to health professionals.

During our interviews, the questions asked to medical staff centered around understanding the challenges and complexities of managing systemic radionuclide therapy (SRT), specifically with Lutathera. The interviews revealed a strong focus on risk management strategies across hospital protocols, radiation safety, and their training. A key theme was the multidisciplinary approach in SRT, involving different roles. All interviews have been recorded. To safeguard the quality and comparability of the interviews, a questionnaire and protocol to summarize the interviews was developed. Interviews with healthcare professionals and researchers were summarized according to this protocol. The interviews with patients and a patient representative have been fully transcribed and translated from French to English. The interviews were analyzed by qualitative thematic analysis based on our research questions, as well as findings from a literature review and the exploratory interviews. All interviewees signed an informed consent. The research was evaluated by the ethics commission of [authors institution].

## Results

### Approval and implementation

The regulation of risk in healthcare, and especially in innovation, is embodied in protocolization, processes, and guidelines. In $^{177}$Lu, this happens at different scales. The already

mentioned approval of [177]Lu by the FDA was a milestone in the history of NETs management and the evaluation of risks involved [15,25].

In order to implement apply [177]Lu, hospitals are required to demonstrate to the relevant French authority, the ASN –Autorité de Sécurité Nucléaire, the French nuclear safety authority, and the manufacturer, their capacities and logistics to handle the involved risks. This means providing proof of the expected benefits of the treatment and their measures to handle risk. Not all nuclear medicine departments are eligible to implement [177]Lu since they must meet certain prerequisites. Processes of getting approval involve considerable administrative effort. Furthermore, approval of [177]Lu is managed at national levels and is not harmonized across Europe, as one nuclear physician adds *"There is no European organization, it's a pity."*

French authorities did not authorize the treatments using [177]Lu when they got implemented in other European countries. By this time, some French patients were sent abroad, and their treatment costs were reimbursed by national social security. Health professionals, such as a radio pharmacist, highlight national differences in terms of regulation in Europe and the administrative burden of getting approval in France which cause delay availability:

"In France, it is necessary to obtain authorization from several authorities, in particular from ANSM1 [Agence nationale de sécurité du médicament, which is the French National Drug Safety Agency] we have regulations that are quite heavy in my opinion, but that's fine, it's still radioactive medicine, but yes, it's extremely heavy. I don't know enough about the regulations elsewhere, but I have the impression that in Germany it's much simpler. I have the impression that in Italy it is also much simpler. So why is it more complicated? Because it takes time, there are big files to fill in, and I have the impression that we always get the authorizations a little bit after the others."

Approval for this treatment not only concerns various responsible authorities but patients as well. As with all medical intervention, treatment requires patients' Informed Consent (IC) as part of the protocol. IC is a fundamental ethical principle that should provide patients with a clear understanding of the benefits and risks, and thus increase legal certainty on the side of the medical agent. It informs patients about the process, risks, and radiation safety measures. The procedural values of accountability, transparency, and inclusiveness are named as relevant for long-term diseases such as NETs [8]. It also provides legal certainty for hospitals and medical personnel.

## Dosimetry and imaging to monitor risks

Radioactivity is monitored during treatment to manage risks and assess effectiveness. Dosimetry and imaging personalize doses and ensure safety for patients and healthcare workers. The aim is to integrate diagnostics with therapy for more individualized treatments. [26]. Dosimetry offers a particular, quantified, and instantaneous expression of risks and benefits of the treatment: the body exposed to radioactivity is monitored. This approach translates the effects of radioactivity on the body by generating data, signs, and symbols that the health professionals seize upon to describe the situation and prescribe actions. Health professionals constantly refer to the data produced, as a medical physicist describes:

"There are very interesting alarms. There are instantaneous dose rate alarms, that is, if I enter a lab and there is a radioactive source, as in a radio pharmacy, the alarm will sound, indicating that there is a radioactive source, and there is a very high dose activity. And also alarms of maximum daily dose rate and so on."

Dosimetry and imaging both express and impose thresholds for the acceptable level of radiation exposure. They constitute tools around which a whole monitoring and action management system is organized. As a care assistant recounts, the ringing of the dosimeter "*reminds us of our orders*" and leads to the adoption of radiation protection measures when thresholds are exceeded. This expression characterizes the trust that health professionals have in medical devices to control risks throughout the treatment.

## Training and expertise

Training and experience of health care professionals is named in literature and interviews as fundamental factor to implement [177]Lu. Health care professionals learn from other hospitals and colleagues who are experienced in administering the treatment. A nuclear physician explained how he learned from more experienced hospitals and how these also took up French patients:

"So, I contacted a pioneer center, the Rotterdam center, so the K. team, they're heavyweights in everything to do with neuroendocrine tumors. […] So, I made contacts and started sending patients to Rotterdam, as one of the doctors and nurses in Rotterdam spoke French. And then I also sent several patients to Austria, to Professor V. because the French doctor and nurse in Rotterdam were no longer there, so the language was starting to pose a problem for patients."

The application of 177Lu requires specific steps and precautions to be implemented and performed in hospitals [9]. Training can also be internal: a radio manipulator, indicated his training of new team members:

"I'm in charge of training newcomers to the RIV [Vectorized internal radiotherapy], I do the procedures with the other two people. They do what I do. We all do the same thing. We're starting to train more and more to do the treatment."

The limited number of patients with NETs eligible for [177]Lu justifies why only a limited number of health professionals have developed expertise in the administration of this treatment. Their role is to "*be*[ing] *trained enhancing public understanding of radiation risk*" [27]. Regular safety training is mentioned as one key aspect of reducing radiation risks [27,28]. Training mainly has a preventive role and requires continuous consideration of radiation protection rules and the management of risks associated with the exposure to radioactivity [29]. This includes various manipulations of the radioactive product to ensure that no radioactive element escapes before injection into the patient's body. A team manager explained that training of staff was necessary to learn handling the radioactive substance:

"With the manipulators, who were not used to working with such high levels of radioactivity, in any case in liquid form, it was necessary to use a system that would allow the product to be injected into the body. […] So, it required a minimum of attention, of concentration, because once it's [the product vial] pierced, it's done. And not to do other things, because otherwise, you're going to expose yourself, you risk getting contaminated. It requires a certain vigilance."

In addition to receiving training, and acquiring know-how in risk management, health professionals learn to anticipate risks that might lead to contamination, such as a leak during the injection, or improper radioprotection protocols that exposes health professionals. They

are also accustomed to managing the day-to-day operations of the department due to frequent *"incidents"*, as the radiation protection manager attests. This requires vigilance and decisions such as close-down of a department or a room if radioactive leakage occurs. Carrying out treatment requires a specific setting, skills, and management of radioactivity both at hospital level but also on practice levels of individual health care professionals. Risk management is present during the entire treatment, from the preparation of the vials that contain the radioactive product to patient management, the injection phase and its aftermath. A radiology technician explained that, during injection, care is required to avoid contamination.

> "There is a specific set-up to be done, you have to respect the procedures because the main risk is the leakage, so it's not trivial, if you put radioactivity everywhere you're in trouble."

[177]Lu implementation requires the capacity of hospital teams and settings to face different levels of risks: for patients given their health state, and for health professionals who are exposed to radioactivity, either from the drug itself or the patients once they have received the injection.

### Hospital logistics and settings

[177]Lu poses logistical and organizational challenges which also include the adaptation of the hospital structure.

Fig 2 represents the spatial organization of a brachytherapy department in a French hospital, with specific zones designated for different levels of radioactive exposure to ensure the safety of patients, healthcare professionals, and visitors. The department is divided into "hot" and "cold" zones, marked by color codes to indicate radiation exposure levels. The "hot" zones, shown in yellow and orange, are areas where radioactive materials are handled or where patients who have received radiopharmaceuticals are located. These areas require rigorous radiation protection measures, such as protective barriers and restricted access, to

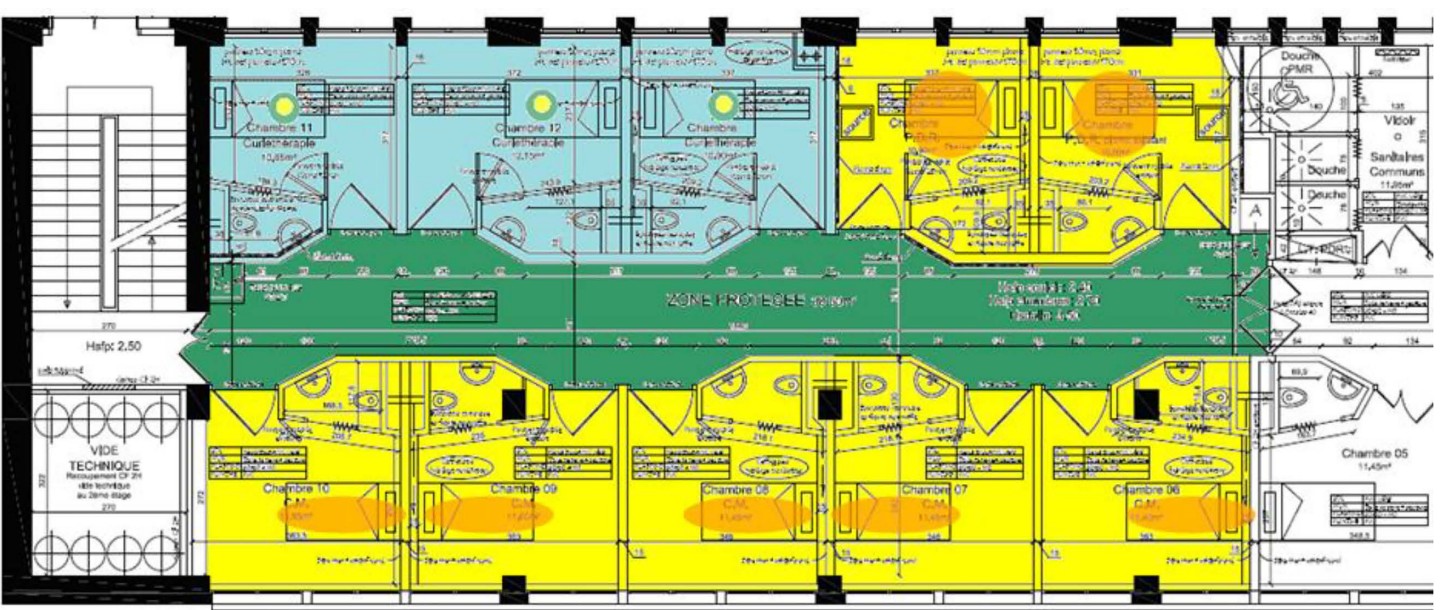

**Fig 2. Zones of a brachytherapy department.**

minimize exposure to healthcare workers and prevent contamination of surrounding spaces. Within these zones, healthcare professionals follow strict protocols, including the use of personal dosimeters and radiation shields, to monitor exposure and ensure adherence to safety standards.

In contrast, "cold" zones, marked in green and blue, are areas with little to no radiation risk, allowing for more standard hospital interactions. These zones serve as transitional spaces where staff and patients can move freely without the need for protective equipment. The physical separation of hot and cold zones involves structural adaptations within the department, such as the installation of walls, lead barriers, and controlled entry points. This layout facilitates a controlled environment that confines radioactive exposure within the hot zones while safeguarding individuals in the cold zones. Fig 2 also includes designated rooms for specific functions, such as patient preparation, radioactive waste storage, and decontamination, with each area tailored to meet regulatory requirements for handling radioactive materials. This spatial organization not only ensures compliance with radiation safety protocols but also enhances operational efficiency by clearly delineating spaces according to radiation risk. Overall, Fig 2 demonstrates the comprehensive infrastructure required in a brachytherapy department to balance effective treatment delivery with rigorous safety standards for radiation protection.

Hospital settings are designed to contain radioactivity in many different ways throughout treatment. This affects how hospital care is organized. This risk arises from handling radiopharmaceuticals but also from working with, and in the presence of patients once they have received radiopharmaceuticals. After drug injection, radioactive substances flow through the patients' bodies for several hours. During that time, patients' bodies become irradiated. Their bodies are considered radioactive until the radioactivity decreases, and radioactive compounds are excreted. The management of urine containing radioactive material, its reception, and storage requires special attention, as a radiation protection manager explained:

> "What we're most interested in (…) is urine. We have said, that out of 7.4 giga-becquerels that we will inject into the patient, 7 giga-becquerels will go into the urine and will leave directly. So, we have to collect the urine. This can only be done in a radio-protected room. There is only this place in the hospital where the toilets are connected to the tanks. It's quite an adventure, you can't imagine!"

According to a radio pharmacist these requirements also pose certain limits to the numbers of patients that can be treated.

> "If we could inject the patient and send them home without question, we would do much more. What prevents us from doing so is the waste, in fact, especially the urine."

Safety measures include also radioactive waste management to avoid contamination [8]. This includes containing radioactive waste after treatment, which is an important logistical prerequisite to implement the treatment. Medical waste management ensures the safety of patients and health professionals and environmental protection. Everything that gets in contact with the patients' bodies during treatment and thereafter, such as linen and food scraps, is considered contaminated and subject to radiotoxicological analysis, as a radio protection manager explained:

> "All the waste here is kept. Even the leftovers from meals are kept. Nothing goes into the garbage, not the laundry, nothing at all."

Health professionals explained how they minimize risks by keeping patients in the nuclear medicine department during treatment, avoiding the need for transportation. As one endocrinologist noted, patients remain in the hospital until their radioactive emissions are low enough for them to safely leave the facility:

"we keep the patient on an almost exclusive basis of radioprotection and irradiation to the people around the patient."

In other words, the irradiating power of the patient's body and the risk their discharge form hospital entails to people they might have contact with determines how long patients stay in the radio protected room.

## Radiation management throughout treatment

Regulations, protocols and organizational principles provide a framework for health professionals and patients how to act and which steps to follow. To follow the safety protocols is seen as critical to manage risks and to implement and ensure successful treatment.

The safety protocols set up in hospitals requires involved personnel to be trained to know the steps and different measures to be taken during the administration of radioactive substances. Since the patients are both object of care and a potential source of risk, they must be involved in the processes of risk management. Indeed, health professionals and contact persons of patients are also confronted with the potentiality of radioactive contamination. Novartis, the manufacturer of $^{177}$Lu, warns patients and health professionals about radiation exposure:

"Treatment with LUTATHERA contributes to a patient's overall long-term cumulative radiation exposure and is associated with an increased risk for cancer. Radiation can be detected in the urine for up to 30 days following LUTATHERA administration. Minimize radiation exposure to patients, medical personnel, and household contacts during and after treatment with LUTATHERA consistent with institutional good radiation safety practices, patient management procedures, Nuclear Regulatory Commission patient release guidance, and instructions to the patient for follow-up radiation protection at home." [30]

Those who handle the radioactive source before treatment, such as radio pharmacists and radiology manipulators are directly exposed to the product's radiation, as are those involved in injecting $^{177}$Lu during the treatment. A radiology technician, who works in brachytherapy, explained how her presence throughout the transfusion is required for about 30 minutes. A nuclear physician underlined the importance of the radioactivity to which the patient and she are exposed:

"I don't know, maybe 5,000 mega becquerels that are passing through the patient, so the patient receives their treatment, so they become more and more radioactive."

This exposure is observed throughout the treatment by the medical physicists who, carry out exposure rates for the medical staff. For this reason, these health professionals wear dosimeters that evaluate their exposure in real-time. Several health professionals have expressed their concerns and perceptions of risk. A nurse assistant had felt the effects of radiation and discomfort even though her dosimeter did not indicate a sufficiently high exposure to be concerned about it:

"When I entered the irratherapy rooms I had a headache. […] At first, I thought it was me who was getting ideas, but it was not. Especially because it is a particular room, because

at the time there was volatile treatment, and here it is ventilated by systems of airings. I still had pain. Now I'm used to it I think, but at the beginning I had headaches. As we are controlled and we don't exceed the dose, I took some doliprane [a drug used to relieve mild to moderate pain and fever.] and took it with me. Anyway, we are controlled every year. There is a truck that comes to give us a dosimetry test [...]. Last year, they did a whole body and thyroid test. I didn't have anything at all.

The management of radioactivity is also a responsibility ascribed to patients. They must follow radiation protection measures, to protect individuals they might expose – including the health professionals and to reduce the effects on their bodies.

The process of managing risks is supported by a specific division of labor, in which each professional involved plays a role [31]. Since this treatment is practice-oriented and emphasizes the need to involve multiple actors in radiation protection, it is multidisciplinary. This implies that the tasks and roles are divided between the different healthcare professionals involved. It includes, for example, nursing assistants, nurses, radiology manipulators, and nuclear or endocrinology physicians. Other professionals are mobilized in the implementation of the protocol as well, its coordination, its efficiencies, and its safety, such as the team manager, the person in charge of radiation protection, and the medical physicist who considers the hospital's entire environment.

Administration of $^{177}$Lu requires health professionals and patients' working together to ensure the safety of everyone. The necessity of managing risks therefore implies organizational and interpersonal trust to foster collaboration between both health professionals and patients.

## The significance of the relationship between health care professionals and patients

Health professionals emphasized the necessity to account for the specific context of each patient. Trust develops through patients' perception that health professionals act in their best interests, which is fostered when they feel their unique circumstances are genuinely understood and respected. This mutual understanding is essential, as health professionals, like the radiology technician quoted below, emphasized the importance of responsibility related to their work:

> "It's a pretty big responsibility, given that these are high-dose treatments, so one dose can cause the patient's death, so it's complicated."

The responsibility expressed by health professionals is based on anticipation, evaluation, and controlling of risks. During the first consultation with the patients, physicians must mention the different risks patients will be exposed to. For example, $^{177}$Lu affects in particular the kidneys. The injection of intravenous amino acids protects them and promotes elimination of the radioactive material. Treatment also involves a minor risk of acquiring leukemia. Health care professionals are expected to communicate to the patients the effects of radiation and how to limit undesirable effects; before each injection they monitor patients' health, measure the effects of radioactivity on their bodies, and inform them about the protocol followed. An endocrinologist explained:

> "There is also something else that is important to tell them, and that is that there is a risk of leukemia, between 1 and 2%. It's mandatory to tell them this at the consultation, and I know that not all doctors do it, because it's difficult to explain to someone anxious [...] And that we will stay away from you because the treatment is radioactive. So, it's psychologically difficult. And on top of that, they are told they have a very low risk of blood disease, including leukemia."

Thus, communication of risks, as well as their moral and psychological consequences for the patient is a central aspect in the consultation. This openness about the potential risks of the treatments is described, by professionals and patients, as key for creating and fostering trust between the physician and the patient. Usually after the first injection and experience in the brachytherapy department, patients start feeling more at ease in this environment. One patient, after having received all cycles of treatment, recalled the initial apprehensions he felt, and how his perception evolved throughout the treatment:

> "I didn't know anything about it. Dr. D. had told me that it wasn't trivial either, I don't remember exactly how she introduced me to it. So, I told myself it might not be easy, but I'm ready to play the game. So yes, I may have been a little apprehensive, but I told myself that it would work. But, apart from the fact that I had to put two catheters in my arm, and a little nausea when they gave me the treatment, there was nothing really bad. So, once I'd had the first one [dose] and seen how it went, I'd come in with my hands in my pockets, and no anxiety at all."

This recount of a patient shows that trust in treatment is a process in which the patient-physicist relationship, and getting accustomed with the treatment, play a role.

Throughout the treatment, the perceived quality of care goes hand in hand with the development of a relationship of trust between health professionals and patients. Exposure to risk, and consenting to it, not only requires confidence in technical knowledge and treatment protocolization. Trust unfolds in the therapeutic relationship. As Origgi [31] points out, the verb *confide²* means to hand over something precious to someone, trusting in them and thus surrendering to their benevolence. As part of this treatment, patients express their consent during the consultation to undergo a clinical examination and accept the treatment. It appears that their perception of the benefits versus the risks involved is intrinsically linked to their trust in their doctor. This relationship is shaped all through the different stages of the treatment. Indeed, in a few cases of patients, the prospect of radiation can create apprehension and even refusal, as encountered by nuclear physicians interviewed. The first stage, the preliminary consultation, highlights the representations of radioactivity and the perception of risk, which requires reassurance for patients, according to a nuclear physician:

> "We see Chernobyl when we hear nuclear medicine. So, I have to say that since I've been here, I think I've had three patients who afterward refused. At this point [of the consultation], I don't get much feedback, because if the patient doesn't consent at the end of this consultation, which I understand, I give them a period of reflection of let's say two to three weeks where they can reflect and discuss in their familiar environment. And then, we contact them again, or they contact us, to say yes or no. And if it's no, I usually don't call back to ask why."

Within the framework of the treatment, trust fosters a progressive trivialization of risk as well, for health professionals and the patients who rely on them. The cases of treatment-refusals mentioned by this radio pharmacist, underlines how perceived risks and associations, also with nuclear bombs and power plant incidents, influence patient's decisions [31,32]. In other words, representations of the dangers and risks of the treatment are informed by the informational context that constructs them. The quality of the relationship between health professionals and patients is critical for the patients to accept being exposed to radiation. This was emphasized by both, health professionals and patients. Yet, associated effects of radioactivity can also underline the efficacy of the treatment, as a nuclear physician said:

> "Especially since I think it makes them feel good the idea of imagining this tumor destroyed by radioactivity"

## Discussion

Risk is a central part of our lives today. Sociologist Ulrich Beck [33] emphasizes the anxieties and insecurities that define the modern age and perceives risk as a key explanatory to account for the multifaceted and all-encompassing (technological) transformation in work, relationships, and politics of modernity. Similarly, Anthony Giddens puts risk assessment center stage in his analysis of modernity and, what he calls, the *"colonization of the future"* [34]. François Ewald, e.g., by focusing on work-related illnesses at the end of the 19th century [35], shows how the concept of risk is connected to the rise of modern *"insurance societies"*, which requires quantification of hazards to render them manageable. Thus, risk became an instrument for the government of hazards which rests on diverse social, technological, as well as informational tools. However, it is insufficient to conceptualize risk on probability and based on (Western) rationality only. Anthropologist Mary Douglas emphasizes the social construction of risk and shows its moral underpinning and relationship with blame [36]. Based on her work, several studies have tried to articulate these theories by analyzing the tools and techniques for managing risks, and how they shape social interactions [37,38].

We have shown different notions and aspects relevant for risk in this paper. Medical devices constitute one central socio-technical tool for evaluating and dealing with risks. They are *"semiotic technologies"* [39] as producers of meanings and are central to evaluating the risks taken during each stage of the treatment for both the patients and health professionals. In addition to ensuring safety, they produce agency [40]. They measure the effects of the exposure of bodies and make their regulation, surveillance, and disciplining them possible. Management of risk is an essential part of the treatment. The ability to deal with unforeseen events and guarantee safety even in cases in which radioactivity is no longer under control, and to recognize the signs of excessive radiation, is as important as other steps of the treatment. Despite their vigilance and rigor during administering the treatment, health professionals cannot ensure that radioactivity is always under control, and incidents must be considered. A radio pharmacist explained the importance of ensuring anticipation, *"all the potential and imaginable risks, even the craziest ones, on the radioactive products."* Potential risks have to be anticipated because they could disrupt the course of treatment. Even if physicians try to foresee risks as completely as possible, it is not possible to control everything that could eventually happen. To be able to deal with the hazards associated with the treatment, complex protocols, and trained professionals are, required as well.

This research has shown that the management of $^{177}$Lu not only requires trust in medical devices, technical skills of health care professionals, organizational settings, and processes, but also interpersonal trust to overcome anxieties and fears associated to radioactivity. From a system-theory perspective, trust can be seen as a way to reduce complexity among interdependent individuals. As a consequence, trust has to be distinguished from familiarity, which characterizes unreflective bounds in everyday life [41]. Thus, trust is also a kind of promise that enables actors to experiment with new modes of action. Several studies have shown how trust is fundamental to ensuring quality healthcare [42–44], enabling action, cooperation, and knowledge sharing.

Our research emphasizes that trust operates at several levels: in the interaction between humans and medical devices, in the cooperation between different health professionals in specific healthcare organizations characterized by an important division of labor, but also between health professionals and patients, who both can be negatively affected by radioactivity. This characteristic of the treatment is quite specific since both patients and health professionals are trusters and trustees, and health professionals have to trust their patients as much as the latter have to trust them to manage risks. This creates a *"community of risk"* [45], in which both parties are vulnerable and share a common condition in front of uncertainties that

go beyond the quantification of foreseeable hazards. This shared condition fosters cooperation of health care professionals and patients concerning the management of risks. In contrast to other treatments, such as chemotherapy mentioned during our fieldwork, patients' concerns and experiences do not reveal a significant gap between the patient's point of view and that of the medical and nursing staff exposed. Taking into account collective exposure to risk is as much a question of medical technique as it is the expression of a new patient-health professional cooperation. Patients and medical staff are required to exercise a new skill which is risk management, not only carried by health professionals. They are both players subject to demanding constraints (understanding the nature of risks) and standardization (adhering to and following elaborate guidelines).

As a consequence, our research shows the importance of considering the process of creating and fostering trust throughout the treatment to deal with risks at three intertwined levels: (1) epistemic trust, which is based on medical evidence by the manipulation and use of medical devices and a constant processes of quantification of calculable risk, as well as the capacity of health professionals to manage them; (2) organizational trust through the definition and implementation of protocols and a division of labor and cooperation between highly trained health professionals, the patients and through the physical organization of the hospitals, and (3) interpersonal trust through effective communication between health professionals and patients throughout treatment to create a shared understanding and representation of the therapy. The quality of communication is associated with the empathetic attitude of healthcare professionals when they are understanding, non-judgmental interlocutors, and when they take into consideration the individual situation of each patient. Our research emphasizes the importance of the intertwinement between these three levels of epistemic, organizational, and interpersonal dimensions of trust to dealt with risks associated with radioactivity. They are necessary to manage risks associated with a complex, innovative treatment.

Our results thus substantiate previous international studies that have conceptualized trust in the context of healthcare organizations, and that have shown that trust relies on both rational, instrumental, and bureaucratic features on the one hand, and interpersonal communication to build consensus and a shared understanding of the situation on the other hand [46,47]. This research thus emphasizes the need to better conceptualize the role of trust in the management of risky medical procedures in various hospital settings and different national contexts. This research took place in French hospitals, which raises the question of the influence of the national context on how actors feel, perceive and express trust. France is a country where the governance of healthcare is organized a centralized way, which leads to specific way of managing risk and uncertainty related to healthcare, and increasingly monopolize scientific expertise [48,49]. The implementation, protocols, and healthcare system frameworks in each country leads to differences in how the treatment is administered, implemented, reimbursed, and monitored. In many countries, like France or Italy, the administration of $^{177}$Lu is provided only in specialized centers. Other countries, like Germany, have a less centralized implementation of this treatment [50]. Varying reimbursement processes across Europe affect how quickly and widely $^{177}$Lu was adopted. Moreover, not all European countries had early access programs for $^{177}$Lu, which explains differences in how quickly the treatment was available to patients. French researchers and institutions have played a crucial role in advancing and optimizing the use of $^{177}$Lu [51]. For example, the country's Temporary Authorization provided early access to $^{177}$Lu, which aided its implementation in local hospitals. However, this process was challenging, particularly in managing the associated risks.

A quantitative survey, operationalizing the concept of trust and distinguishing between the three dimensions that we distinguished, could help better understanding the different dimensions of national healthcare systems – organization and size of hospitals, levels of

centralization and bureaucratization of institutions, healthcare insurance, number of patients for one clinician – which foster or jeopardize levels of trust in different hospitals and in different countries, both for treatments associated to radioactivity and other kind of risks.

## Conclusion

In this paper, we analyzed health professionals' and patients' experiences of Systemic Radionuclide Therapy (SRT) using $^{177}$Lu in France, based on qualitative interviews and ethnographic observation. Our research highlights how risk is dealt with and managed in hospitals to enable the implementation of SRT. The articulation of risk management and provision of care leads to a specific organization of the treatment and requires the combination of multiple socio-technical devices, settings, practices, monitoring, and specific cooperation to manage risk. However, this article argues that to manage risk, fostering different, multiple dimensions of trust between the different actors involved is a necessity. Trust is not only built by technical and organizational competence and skills but also by giving the patients a feeling that they are taken care of that their concerns and needs are taken seriously and that they are listened to, in order to build a shared representation of the treatment and of its risks. This sense of being heard has been crucial for patients to feel secure in their care journey. By actively engaging in dialogue, health professionals fostered a shared representation of the treatment and its risks, ensuring that patients feel informed throughout the process. By cultivating this collaborative environment, trust was deepened, leading to better patient outcomes and satisfaction. Therefore, the quality of the relationship between health care professionals and patients is a prerequisite to foster trust in therapeutic innovation and to manage risks. Implementing SRT calls for a reflection on the concept of risk in healthcare, by reminding us of its social and interpersonal foundations and going beyond its cognitive and quantifiable dimensions.

## Acknowledgments

We acknowledge the support and participation of patients and health professionals from the Centre Léon Bérard, Hôpital de la Pitié-Salpêtrière, Institut de Recherche en Cancérologie de Montpellier, Institut Gustave Roussy, Centre Hospitalier Universitaire de Dijon, Centre Hospitalier Universitaire de Bordeaux, Hôpital Edouard Herriot, Institut Universitaire du Cancer de Toulouse, Hôpital Haut-Lévêque. We furthermore want to thank Florian Winkler for his contribution to setting up the research protocol and Shauna Stack for language editing.

## Author contributions

**Conceptualization:** Solenn Thircuir, Héloïse Pillayre, Johannes Starkbaum, Erich Griessler.

**Formal analysis:** Solenn Thircuir, Johannes Starkbaum, Erich Griessler.

**Funding acquisition:** Erich Griessler.

**Investigation:** Solenn Thircuir.

**Methodology:** Johannes Starkbaum, Erich Griessler.

**Project administration:** Erich Griessler.

**Supervision:** Erich Griessler.

**Writing – original draft:** Solenn Thircuir, Héloïse Pillayre, Johannes Starkbaum, Erich Griessler.

**Writing – review & editing:** Solenn Thircuir, Héloïse Pillayre, Johannes Starkbaum, Erich Griessler.

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
