## [Decision Letter · Decision Letter 0]

5 Aug 2024

PONE-D-23-37206“Taming” Risk  

Perception and Management of Radiation Risks in Cancer Treatment Using 177LuPLOS ONE

Dear Dr. THIRCUIR,

Thank you for submitting your manuscript to PLOS ONE. After careful consideration, we feel that it has merit but does not fully meet PLOS ONE’s publication criteria as it currently stands. Therefore, we invite you to submit a revised version of the manuscript that addresses the points raised during the review process.

 Please submit your revised manuscript by Sep 19 2024 11:59PM. If you will need more time than this to complete your revisions, please reply to this message or contact the journal office at plosone@plos.org . Please include the following items when submitting your revised manuscript:

We look forward to receiving your revised manuscript.

Kind regards,

Hesham M.H. Zakaly, Ph.D.

Academic Editor

PLOS ONE

“This research was funded by ERA PerMed and the Austrian Science Funds (FWF)”

“This research was funded by ERA PerMed and the Austrian Science Funds (FWF). We like to thank our project partners as well as all persons we did interviews with or that helped in getting necessary contacts and institutional access for interviews”

“This research was funded by ERA PerMed and the Austrian Science Funds (FWF)”

5. In the online submission form, you indicated that your data is available only on request from a third party. Please note that your Data Availability Statement is currently missing the name of the third party contact or institution / contact details for the third party, such as an email address or a link to where data requests can be made. Please update your statement with the missing information.

Reviewers' comments:

Reviewer's Responses to Questions

**Comments to the Author**

1. Is the manuscript technically sound, and do the data support the conclusions?

Reviewer #1: No

Reviewer #2: Partly

2. Has the statistical analysis been performed appropriately and rigorously? 

Reviewer #1: N/A

Reviewer #2: N/A

3. Have the authors made all data underlying the findings in their manuscript fully available?

Reviewer #1: Yes

Reviewer #2: No

4. Is the manuscript presented in an intelligible fashion and written in standard English?

Reviewer #1: Yes

Reviewer #2: No

5. Review Comments to the Author

Reviewer #1: Comments for transmission to authors:

Solenn Thircuir and co-authors conducted a qualitative study on managing risk perception and trust in systemic radionuclide therapy (SRT) for cancer treatment, specifically with 177Lu. Their research emphasizes the importance of understanding and fostering trust throughout the treatment process. The article is well-written, and the authors effectively convey their points.

Major points:

1) The manuscript is too long with limited figures. It contains only one figure, which has many details that are not adequately explained in the figure legend. Consider adding more figures, such as those showing the demographics of systemic radionuclide therapy using 177Lu in Europe and worldwide, as well as a pipeline of the treatment process and the roles of the different people involved.

2) The sample size of interviews conducted with healthcare professionals and patients was limited to French hospitals, potentially limiting the broader applicability of the findings. The study's focus on a limited number of participants in France may not fully represent the diversity of experiences with 177Lu treatment. Consider including the country name in the title and abstract.

3) The findings may be specific to the French healthcare system and may not be generalizable to other countries. Could the authors discuss in the discussion section how other developed countries address similar issues, such as waste management and the risks associated with leakage (page 9)?

4) The trust factor highlighted by the authors is very subjective and can vary between locations and individuals. The study focused on a specific type of cancer treatment (SRT with 177Lu) and may not be generalizable to other medical interventions. Could the authors suggest a quantitative method to monitor and universalize the radiation risks associated with 177Lu in the discussion section?

Minor points:

1) Some sentences are in correction mode, please check the following:

Page 13

So,_(twice)

Page 16

_Trust; _Therefore.

2) There is unnecessary capitalization in some sentences.

Page 5

“….thErapeutic protocols with Lu-177 for MNETs, through the development of

advanced computational tools and a portable detection sYstEm”

3) Correct the following:

Page 8

“..manipulator, + “

Reviewer #2: i can see report not scientific paper

Please remove “Taming” Risk  from the title.

The abstract is short and without any results please rewrite.

English is poor and needs English editing.

We show- we will- we applied ..........

The introduction needs reorganization the aim should be at the end not before history

6. PLOS authors have the option to publish the peer review history of their article (what does this mean? ). If published, this will include your full peer review and any attached files.

**Do you want your identity to be public for this peer review?** For information about this choice, including consent withdrawal, please see our Privacy Policy .

Reviewer #1: No

Reviewer #2: **Yes: ** Mostafa Yunees Abdelfatah Mostafa

---

## [Author Response · Author response to Decision Letter 0]

10 Oct 2024

Dear Reviewers,

We sincerely thank you for your thorough reviews and constructive feedback on our manuscript, "Taming Risk Perception and Trust in Systemic Radionuclide Therapy for Cancer Treatment with 177Lu." We have carefully considered all your comments and made revisions accordingly. Below, we address each of your points in detail.

Response to Reviewer 1

1. Technical soundness and data support:

We acknowledge your concern regarding the manuscript's length relative to the limited number of figures. To improve clarity and understanding, we have added additional figures illustrating the demographics of systemic radionuclide therapy using 177Lu in Europe, along with a table highlighting the roles of various stakeholders involved in the treatment process in the methodology section.

2. Sample size and generalizability:

We appreciate your observation about the survey’s focus on French hospitals. To enhance the manuscript’s applicability, we have specified the country name in both the title and abstract. Additionally, we have included a discussion on the potential limitations regarding generalizability and how other Western countries address similar issues, which is now reflected in the revised discussion section.

3. Subjectivity of trust:

In the revised discussion, we have suggested that a quantitative method for monitoring and universalizing the radiation risks associated with 177Lu would further our analysis. This addition aims to foster dialogue and provide a more robust framework for understanding trust dynamics across different medical interventions.

4. Minor points:

In response to your suggestions for minor language editing, we conducted a thorough review of the entire manuscript, correcting misspellings and typos, removing redundant sentences, and ensuring proper capitalization throughout.

Response to Reviewer 2

1. Title and abstract:

We have removed “Taming” Risk? from the title as per your suggestion and replaced it with “Dealing with Radiation Risks in Systemic Cancer Treatment: Perspectives of Practitioners and Patients.” The abstract has been rewritten to include key results and findings from our study, providing a clearer overview of our work and its implications.

2. Language and editing:

To address your concerns about the language, we performed a thorough review of the entire text and made significant edits for clarity. We also engaged a professional language editor to ensure that the manuscript meets high linguistic standards.

3. Data availability:

We would like to clarify that while we aim to adhere to data availability requirements, qualitative data cannot be made fully available due to the sensitivity of personal and health information involved.

4. Introduction reorganization:

The introduction has been reorganized to place the aim of the study at the end, following the historical context. This restructuring improves the flow of information and provides a clearer focus for readers.

In sum, we have thoroughly engaged your suggestions. Thank you for considering our revised article re-submission. We believe that these revisions comprehensively address the reviewers’ concerns and enhance the overall quality of the manuscript. Thank you once again for your valuable feedback. We look forward to your further consideration of our revised submission.

Sincerely,

Solenn Thircuir, Héloïse Pillayre, Johannes Starkbaum, and Erich Griessler

---

## [Decision Letter · Decision Letter 1]

28 Oct 2024

PONE-D-23-37206R1Dealing with Radiation Risks in Systemic Cancer Treatment

Perspectives of practitioners and patients in French hospitalsPLOS ONE

Dear Dr. THIRCUIR,

Thank you for submitting your manuscript to PLOS ONE. After careful consideration, we feel that it has merit but does not fully meet PLOS ONE’s publication criteria as it currently stands. Therefore, we invite you to submit a revised version of the manuscript that addresses the points raised during the review process.

 Please submit your revised manuscript by Dec 12 2024 11:59PM. If you will need more time than this to complete your revisions, please reply to this message or contact the journal office at plosone@plos.org . Please include the following items when submitting your revised manuscript:

We look forward to receiving your revised manuscript.

Kind regards,

Hesham M.H. Zakaly, Ph.D.

Academic Editor

PLOS ONE

Journal Requirements:

Reviewers' comments:

Reviewer's Responses to Questions

**Comments to the Author**

1. If the authors have adequately addressed your comments raised in a previous round of review and you feel that this manuscript is now acceptable for publication, you may indicate that here to bypass the “Comments to the Author” section, enter your conflict of interest statement in the “Confidential to Editor” section, and submit your "Accept" recommendation.

Reviewer #1: (No Response)

Reviewer #2: All comments have been addressed

2. Is the manuscript technically sound, and do the data support the conclusions?

Reviewer #1: (No Response)

Reviewer #2: Yes

3. Has the statistical analysis been performed appropriately and rigorously? 

Reviewer #1: (No Response)

Reviewer #2: N/A

4. Have the authors made all data underlying the findings in their manuscript fully available?

Reviewer #1: (No Response)

Reviewer #2: Yes

5. Is the manuscript presented in an intelligible fashion and written in standard English?

Reviewer #1: Yes

Reviewer #2: Yes

6. Review Comments to the Author

Reviewer #1: 1) Authors should address each reviewer comment individually, rather than providing general responses about the changes made.

For example:

Reviewer Comment 1:

Author's Response:

2) Replace 'graph 1' with 'figure 1' and add a legend for figure 1. Additionally, rename the current figure 1 to figure 2

3) The previous comment regarding the figure 1 legend has not been addressed. The current diagram of the brachytherapy department is too complex. Please either elaborate on the figure 1 legend, even if the information is also provided in the results, or simplify figure 1 itself.

Ensure the format follows conventional standards.

For example:

Figure 1: Zones of a brachytherapy department. Add detailed description here.

4) Please review the title to ensure it includes any necessary punctuation marks that might be missing.

5) Replace “pathway” with “pipeline”.

6) Double-check punctuation throughout the manuscript, particularly in newly modified text.

Reviewer #2: Authors address all comments as reviewers suggested. The revised manuscript can be accepted for publication

7. PLOS authors have the option to publish the peer review history of their article (what does this mean? ). If published, this will include your full peer review and any attached files.

**Do you want your identity to be public for this peer review?** For information about this choice, including consent withdrawal, please see our Privacy Policy .

Reviewer #1: No

Reviewer #2: **Yes: ** Mostafa Yuness Abdelfatah Mostafa

---

## [Author Response · Author response to Decision Letter 1]

19 Nov 2024

Dear Reviewers,

We sincerely thank you for your thoughtful and constructive feedback on our manuscript, "Dealing with Radiation Risks in Systemic Cancer Treatment: Perspectives of Practitioners and Patients in French Hospitals." Your detailed suggestions have been invaluable in refining and improving our work. Below, we provide a point-by-point response to the comments raised, addressing all feedback from both of you.

Reviewer 1

Comment: Replace 'graph 1' with 'figure 1' and add a legend for figure 1. Additionally, rename the current figure 1 to figure 2.

Response:

Page 4: 'Graph 1' has been replaced with 'Figure 1.'

Page 10: 'Figure 1' has been renamed as 'Figure 2.'

A legend and description were added to Figure 1 (Page 4), and its title was reformulated:

Figure 1: 177Lu Treatment Pipeline and Involvement of Healthcare Professionals.

Comment: The previous comment regarding the figure 1 legend has not been addressed. The current diagram of the brachytherapy department is too complex. Please either elaborate on the figure 1 legend, even if the information is also provided in the results, or simplify figure 1 itself. Ensure the format follows conventional standards.

Response:

A detailed description was added to Figure 2 (Page 11).

We have revised the format to align with conventional standards.

Comment: Please review the title to ensure it includes any necessary punctuation marks that might be missing.

Response:

Titles throughout the manuscript have been reviewed and corrected for punctuation.

Comment: Replace “pathway” with “pipeline.”

Response:

“Pathway” has been replaced with “pipeline” (Page 4).

Comment: Double-check punctuation throughout the manuscript, particularly in newly modified text.

Response:

A thorough punctuation check has been completed across the manuscript.

Reviewer 2

We deeply appreciate your acknowledgment that the revisions from the previous review round have adequately addressed your concerns.

Comment: The revised manuscript meets the expectations for clarity and completeness.

Response:

Thank you for your positive assessment and support.

Additional Feedback: We encourage careful proofreading for consistency in formatting throughout the manuscript.

Response:

The manuscript has undergone a comprehensive review for consistency in formatting, including figures, tables, and references.

Closing Remarks

We believe that these revisions comprehensively address all points raised and significantly enhance the manuscript's clarity and quality. Thank you once again for your valuable feedback and for considering our revised submission. We look forward to your further comments or confirmation of readiness for publication.

Sincerely,

Solenn Thircuir, Héloïse Pillayre, Johannes Starkbaum, and Erich Griessler

---

## [Editor Report · Decision Letter 2]

20 Dec 2024

Dealing with Radiation Risks in Systemic Cancer Treatment

Perspectives of practitioners and patients in French hospitals

PONE-D-23-37206R2

Dear Dr. THIRCUIR,

We’re pleased to inform you that your manuscript has been judged scientifically suitable for publication and will be formally accepted for publication once it meets all outstanding technical requirements.

Kind regards,

Hesham M.H. Zakaly, Ph.D.

Academic Editor

PLOS ONE
---

## [Editor Report · Acceptance letter]

PONE-D-23-37206R2

PLOS ONE

Dear Dr. Thircuir,

I'm pleased to inform you that your manuscript has been deemed suitable for publication in PLOS ONE. Congratulations! Your manuscript is now being handed over to our production team.

Kind regards,

on behalf of

Dr. Hesham M.H. Zakaly

Academic Editor

PLOS ONE